# Efficient Knowledge Transfer in Federated Recommendation for Joint Venture Ecosystem

**Yichen Li**[1], **Yijing Shan**[1], **Yi Liu**[2] **Haozhao Wang**[1], **Cheng Wang**[1],
**Wei Wang**[2], **Yi Wang**[2], **Ruixuan Li**[1]*

[1]School of Computer Science and Technology, Huazhong University of Science and Technology,
Wuhan, China [2]Chongqing Ant Consumer Finance Co., Ltd, Ant Group, Chongqing, China
`{ycli0204,M202474082,hz_wang}@hust.edu.cn`

## Abstract

The current Federated Recommendation System (FedRS) focuses on personalized recommendation services and assumes clients are personalized IoT devices (e.g., mobile phones). In this paper, we deeply dive into new but practical FedRS applications within the joint venture ecosystem. Subsidiaries engage as participants with their users and items. However, in such a situation, merely exchanging item embedding is insufficient, as user bases always exhibit both overlaps and exclusive segments, demonstrating the complexity of user information. Meanwhile, directly uploading user information is a violation of privacy and unacceptable. To tackle the above challenges, we propose an efficient and privacy-enhanced Federated Recommendation for the Joint Venture Ecosystem (FR-JVE) that each client transfers more common knowledge from other clients with a distilled user's *rating preference* from the local dataset. More specifically, we first transform the local data into a new format and apply model inversion techniques to distill the rating preference with frozen user gradients before the federated training. Then, a bridge function is employed on each client side to align the local rating preference and aggregated global preference in a privacy-friendly manner. Finally, each client matches similar users to make a better prediction for overlapped users. From a theoretical perspective, we analyze how effectively FR-JVE can guarantee user privacy. Empirically, we show that FR-JVE achieves superior performance compared to state-of-the-art methods.

## 1 Introduction

Recommendation systems have emerged as crucial tools and products, significantly impacting daily lives by offering tailored suggestions of new items that may interest users. These systems fundamentally rely on centralized servers to consolidate user data, digital behaviors, and preferences, thereby training models for precise recommendation generation [9, 39, 8]. However, transmitting local user information to central servers poses considerable privacy and security concerns. Moreover, recent stringent government regulations on privacy protection underscore the necessity of storing user data locally on devices rather than uploading it to a global server. As a potential remedy to this dilemma, federated learning (FL) emerges as a promising approach, enforcing data localization and enabling the distributed training of a globally shared model [18, 12, 37, 13]. This framework has achieved remarkable success and has been applied to various fields, such as recommendation systems [38, 27, 14] and smart healthcare [6, 34].

In recent years, researchers have studied federated recommendation systems (FedRS), which enable different clients to yield optimal recommendations without breaching user privacy. FCF proposed

---

*Ruixuan Li is the corresponding author.

39th Conference on Neural Information Processing Systems (NeurIPS 2025).

in [1] is the first FL-based collaborative filtering method, which employs the stochastic gradient approach to update the local model, and FedAvg is adopted to update the global model. The authors in [4] propose to adapt distributed matrix factorization to the FL setting and introduce the homomorphic encryption technique on gradients before uploading to the server. FedNCF [10] adapts neural collaborative filtering to the federated setting which introduces neural network to learn user-item interaction function to enhance model learning ability. To better protect user privacy, the authors in [27] offer a bipartite personalization mechanism for personalized recommendations via keeping the user embedding local. Based on it, FedRAP [21] further emphasizes the differences between user clients with additive personalization techniques.

While these methods achieve great success in user-level federated recommendation in which the user engages as the participant in the federated training, we focus on another more challenging and realistic scenario of federated recommendation, specifically within a joint venture ecosystem, where each participant is no longer an individual user but rather a subsidiary or business group. These subsidiaries, each possessing their own users and items, are managed under a unified controlling corporation. For instance, Alibaba [23], as the parent company, oversees various businesses such as Alipay [26], Tmall Shopping [11], and Digital Media and Entertainment Group [31]. These business groups need to collaborate while also protecting their respective user privacy information due to potential conflicts of interest. Each group will engage in federated training as a participant, sharing partially overlapping user and item information (e.g., credits, red envelopes, discounts) while maintaining their own exclusive user bases and item offerings. The intuitive explanation can also be found in a simple 3-client example illustrated in Figure 1.

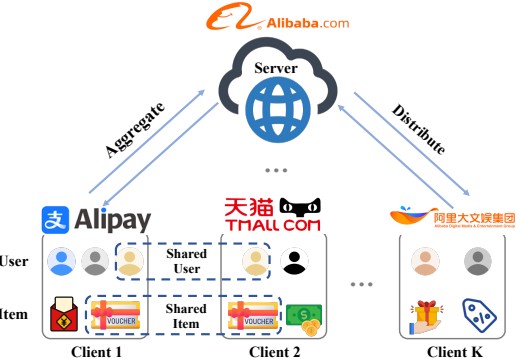

Figure 1: An example of 3-client in the joint venture ecosystem. Each participant in the study represents a distinct business group, encompassing a subset of users and items. Notably, while Client 1 and Client 2 share a partial overlap in their user and item sets, they each maintain a unique set of personalized users and items. The primary objective of FedRS is to leverage collaborative training to develop a comprehensive global model, especially enhancing prediction accuracy specifically for those users who are common across the participating clients.

Compared to traditional federated recommendation scenarios, our ecosystem poses additional challenges. Firstly, there is a significantly larger volume of data involved, as the participants are no longer individual users but rather business groups. Secondly, partial overlap exists between users and items, necessitating selective knowledge sharing among participants to enhance recommendation performance for both shared and unique users. Lastly, there is a trade-off between the privacy of user information and the performance of the recommendation model. Some existing methods upload user embedding to improve model performance, inadvertently leaking private information [10, 4]. Others exclusively utilize item embedding while keeping user embedding locally private [27, 21]. While this approach can yield satisfactory results in traditional federated recommendation settings, in our scenario, where a large data volume is involved, users exhibit partial overlap, resulting in intricate and overlapping user information. Relying solely on the sharing of item embedding falls short of achieving the expected performance.

To address these challenges, we in this paper investigate both a privacy-friendly and efficient method for the joint venture ecosystem. Inspired by [28], most cross-domain recommendation research transfers the common knowledge in the latent space between source and target domains to enhance recommendation performance. To facilitate this transfer, bridge functions are trained to map the embedding of the user and item from the source domain onto the target domain. In our ecosystem, we consider that the local data in each client constitutes a specific domain and study how to more effectively and securely transfer user and item information between clients.

To explore this idea, we first follow the hypothesis of sharing item embedding across clients in a FedAvg manner. While directly sharing user embedding with other clients is unacceptable due to the privacy leakage, we attempt to apply the differential privacy technique [35] to the user embedding to

protect user privacy and employ a bridge function for each client to identify similar users between clients. However, the common understanding of differential privacy is that as the privacy budget decreases, the effectiveness of the model decreases, whereas increasing the budget results in increased privacy risks. Then, we propose an efficient and privacy-enhanced **f**ederated **r**ecommendation framework for the **j**oint **v**enture **e**cosystem dubbed FR-JVE that constructs the rating preference for each client as the filtering signal to transfer the common knowledge from other clients. Specifically, before the federated training, each client will transform the local dataset into a new data format. Based on it, a secure user's rating preference will be constructed locally via the model inversion technique with frozen user gradients, which will not disclose user privacy. Then, the fixed rating preference will be uploaded, and the central server will transmit the aggregated rating preferences of other clients to each client. During the training process, each client only needs to update and communicate with the item embedding. A bridge function is trained locally to map the downloaded aggregated rating preference to the local rating preference. At the inference stage, each client searches the top-k users well-matched on the rating preference mapped by the bridge function for recommendation.

Through extensive experiments on various datasets with rating tasks, we show that FR-JVE significantly improves the recommendation performance compared to state-of-the-art approaches. The major contributions of this paper are summarized as follows:

- We study the problem of federated recommendation for joint venture ecosystem, in which the main challenges are the massive data volume and complex user/item distribution. Different from traditional user-level federated recommendation, the subsidiary or business group with its own users and items participates in the federated training. The balance between performance and privacy does matter here.

- Then, to address this problem, we propose a novel framework named FR-JVE which can be seen as an off-the-shelf personalization add-on for standard item embedding transmission-based federated recommendation system and it involves the user information with a secure rating preference in a privacy-friendly manner to improve the performance.

- Finally, we theoretically show that FR-JVE can efficiently guarantee user privacy and empirically conduct extensive experiments on various datasets. Experimental results illustrate that our proposed model outperforms the state-of-the-art methods on rating tasks.

## 2 Related Work

**Federated Learning** Federated Learning (FL) aims to train a global model collaboratively by aggregating locally trained models from multiple clients, each utilizing its own private dataset [17, 16, 15, 19]. FedAvg [29] is a widely known FL framework that enhances the global model by combining parameters from locally trained models. Methods like [25] employ knowledge distillation with unlabeled samples as a proxy dataset. Recently, data-free knowledge distillation techniques leveraging adversarial methods for data generation have garnered attention [45, 43, 36]. In this manuscript, the FL framework has been adopted to the recommendation system to provide privacy protection for user information and recommendation services.

**Federated Recommendation System** Federated recommendation systems have garnered significant interest lately, fueled by heightened privacy concerns [40, 24, 22]. Recent efforts have primarily concentrated on leveraging the interaction matrix, the cornerstone of basic recommendation scenarios [42, 2]. FCF [1], a pioneering collaborative filtering method under federated learning (FL), utilizes stochastic gradients to update local models and FedAvg for global model aggregation. To protect user privacy, FedMF [4] integrates distributed matrix factorization into the FL framework, encrypting gradients before transmission to the server. Additionally, federated recommendation methods utilizing diverse data sources have emerged, accounting for multiple information streams. FedFast [33] enhances FedAvg with an active aggregation strategy to accelerate convergence, while Efficient-FedRec partitions the model into a server-side news model and a client-side user model, minimizing computational and communication overheads. Both approaches go beyond the interaction matrix, incorporating user features and new attributes. Meanwhile, PFedRec[27] offers a bipartite personalization mechanism for personalized recommendations. Existing research usually focuses on the user-level federated recommendation, where each user is involved as a participant. FedCORE [20] proposes a cross-organization federated recommendation framework for cold-start users. In this

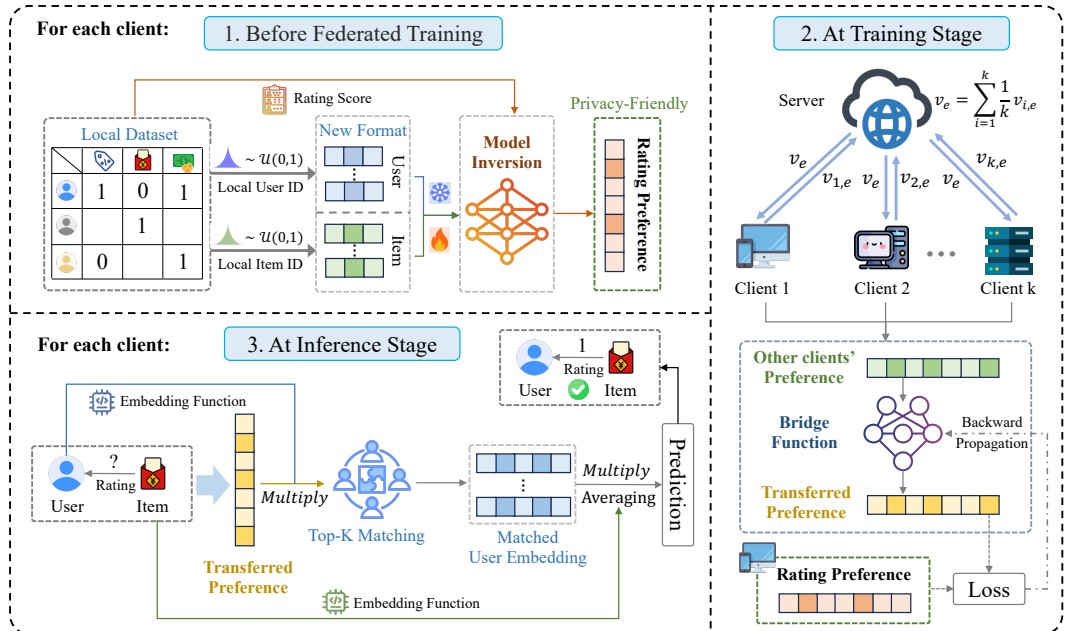

Figure 2: The framework of FR-JVE. Before the federated training, each client first locally transforms the local data into a new data format. Then, the model inversion is applied to distill the privacy-friendly rating preference for each user with the frozen gradient of user information. At the training stage, each client communicates only with the item embedding in a FedAvg manner. A bridge function will be trained to align the local rating preference and other clients' rating preferences. At the inference stage, each client matches similar users to obtain a better prediction.

paper, we study a novel federated recommendation training framework for a joint venture ecosystem and better enhance user privacy while improving recommendations.

## 3 Problem Formulation

**Federated Recommendation.** We aim to collaboratively train a global model for $K$ total clients in FedRS. We consider clients to have a partially shared user set and item set. Each client $k$ can only access to his local private dataset $D_k = \{U_k, V_k, R_k\}$, where $U_k = \{u_k^1, u_k^2, \cdots, u_k^i\}$ has $i$ users, $V_k = \{v_k^1, v_k^2, \cdots, v_k^j\}$ has $j$ items and $R_k = \{r_{11}, r_{12}, \cdots, r_{ij}\}$, $r_{ij} \in \mathbf{R}$ denotes the interaction of user $u_i$ to the item $v_j$. When original local data $\{U_k, V_k, R_k\}$ enters the forward process, matrix factorization will transform it into the representation $(u_k, v_k)\mathbf{E_k} = (u_{k,e}, v_{k,e})$. The global dataset is considered as the composition of all local datasets $D = \{D_1, D_2, \cdots, D_K\} = \sum_{k=1}^K D_k$. The objective of the FedRS is to learn a global embedding model $\mathbf{E}$ that minimizes the total empirical loss over the entire dataset $D$:

$$\min_{\mathbf{E}} \mathcal{L}(\mathbf{E}) := \sum_{k=1}^{K} \frac{|D_k|}{|D|} \mathcal{L}(\mathbf{E_k}). \tag{1}$$

$$\mathcal{L}(\mathbf{E_k}) = \begin{cases} \sum_{(i,j) \in R_k} \frac{1}{D_k}(r_{ij} - \hat{r}_{ij})^2, & \hat{r}_{ij} = [u_{k,e}^i]^T v_{k,e}^j. \;\; ① \\ \sum_{(i,j) \in R_k} \frac{1}{D_k} - (r_{ij} \log \hat{r}_{ij} + (1 - r_{ij}) \log(1 - \hat{r}_{ij})). \;\; ② \end{cases} \tag{2}$$

where $\mathcal{L}(\mathbf{E_k})$ is the loss in the $k$-th client. Here we define two loss functions for ① rating prediction and ② top-k recommendation.

**Cross-Domain Recommendation.** Cross-Domain Recommendation (CDR) focuses on transferring preference knowledge from an auxiliary source domain to a target domain, effectively addressing

the data sparsity issue inherent in conventional recommendation systems. Given the necessity of knowledge transfer in CDR, recent research has predominantly emphasized user embedding and mapping relationships, leveraging latent space bridge functions to transfer user preferences. Suppose that $\{u_{s,e}, v_{s,e}\}$ denotes the user/item embedding of the source domain and $\{u_{t,e}, v_{t,e}\}$ denotes the user embedding of the target domain. Embedding bridge functions $\{\phi_u, \phi_v\}$ are tasked with mapping the user/item embedding from the source domain to the target domain. The learning process is formalized as a supervised regression problem that minimizes the following loss:

$$\min_{\phi_u} \sum_{\bar{u}} \mathcal{L}_{map}(\phi(u_{s,e}), u_{t,e}), \quad \min_{\phi_v} \sum_{\bar{v}} \mathcal{L}_{map}(\phi(v_{s,e}), v_{t,e}). \tag{3}$$

where the $\bar{u}$ denotes the shared users and $\bar{v}$ denotes the shared items, the bridge function is defined with a linear layer, and the loss function $\mathcal{L}_{map}$ with the mean square error loss. During the local inference stage, the final rate is predicted for the user $u_i$ and item $v_j$ by averaging the ratings:

$$\hat{r}_{ij} = \sum_{n=1}^{|\bar{u}|} \frac{1}{|\bar{u}|} [\phi_u(u_{s,e}^i)]^T \phi_v(v_{s,e}^j). \tag{4}$$

where $\hat{r}_{ij}$ denotes the predicted rating for the user $u_{t,e}^i$ and item $v_{t,e}^j$.

## 4 Federated Recommendation for the Joint Venture Ecosystem

In this section, we first analyze the knowledge transfer techniques used in the centralized cross-domain recommendation and examine their application in our ecosystem with partially overlapped user bases and item offerings. We then discuss an important issue: directly uploading the user embedding will breach user privacy. We adopt differential privacy to user embedding, ensuring privacy. Unfortunately, it inevitably results in an unbalance between performance and privacy. Motivated by these findings, we finally present our both efficient and privacy-enhanced FR-JVE method.

### 4.1 Exploration of Cross-Domain Recommendation for Our Ecosystem

In traditional user-based federated recommendation systems, each user participates in federated training to obtain better recommendation results tailored to themselves. In such scenarios, user information is relatively scarce, and promising results can be achieved solely through interactions based on item information. However, in our joint venture ecosystem, where participants act as subsidiaries, there exist vast numbers of users and items. It necessitates federated training to recommend suitable products to all users, where user information plays a pivotal role.

Inspired by [28], the cross-domain recommendation has achieved great success in transferring knowledge from other domains to enhance recommendation performance. Here we try to adopt this technique into the user-based FedRS. Since the client can communicate with the item embedding without breaching user privacy, we just employ a single bridge function to map the user embedding. Suppose that the client $k$ implements an

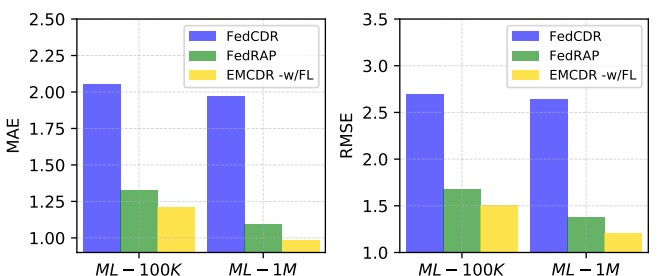

Figure 3: Performance of traditional CDR technique for the joint venture ecosystem.

embedding bridge function $\phi_k$ which maps the local user embedding with user uid. The learning process is formalized as a supervised regression problem that aims to minimize the following loss:

$$\min_{\phi_k} \sum_{u_k^i \in \overline{U}} \mathcal{L}_{map}\bigg(\phi_k(u_{k,t}), u_{k,e}\bigg), \tag{5}$$

$$\text{where } u_{k,t} = \sum_{i=1}^{K/\{k\}} \frac{|D_i|}{|D|/|D_k|} \cdot u_{k,e}.$$

Table 1: Evaluation of differential privacy with the user embedding on two recommendation datasets. Here we underline the budget used in subsequent experiments.

| Dataset | Method | Metric | Privacy Budget $\epsilon$ | | | | |
|---|---|---|---|---|---|---|---|
| | | | $\epsilon = 0.08$ | $\epsilon = 0.1$ | $\epsilon = 0.15$ | $\underline{\epsilon = 0.2}$ | $\epsilon \geq 1.0$ |
| ML-100K | FedMF | MAE | 5.9516 | 5.4994 | 4.9770 | 4.7998 | 3.6019 |
| | | RMSE | 12.977 | 10.6839 | 8.1233 | 7.1237 | 4.3780 |
| | EMCDR -w/FL | MAE | 1.8382 | 1.5983 | 1.3591 | 1.2693 | 1.2123 |
| | | RMSE | 3.5265 | 3.1064 | 2.3998 | 2.0796 | 1.5072 |
| | | | $\epsilon = 0.02$ | $\epsilon = 0.03$ | $\epsilon = 0.04$ | $\underline{\epsilon = 0.05}$ | $\epsilon \geq 0.1$ |
| ML-1M | FedMF | MAE | 5.2194 | 4.8106 | 4.5018 | 4.3850 | 3.6809 |
| | | RMSE | 13.6630 | 10.0847 | 7.8181 | 6.9106 | 4.3610 |
| | EMCDR -w/FL | MAE | 1.4919 | 1.2322 | 1.1329 | 1.0869 | 0.9824 |
| | | RMSE | 4.2600 | 3.2920 | 2.5695 | 2.1763 | 1.2028 |

where the $u_{k,t}$ denotes the global user embedding calculated by weighing other user embedding in a FedAvg manner and $\overline{U}$ denotes the shared user. For simplicity, we define the bridge function with a linear layer and the loss function $\mathcal{L}_{map}$ with the mean square error loss. During the local inference stage, each client makes the final rate prediction for the user $u_k^i$ and item $v_k^j$ by averaging the ratings by similar users:

$$\hat{r}_{ij} = \sum_{k=1}^{|\overline{U}|} \frac{1}{|\overline{U}|} [\phi_k(u_{k,t})]^T v_{k,e}^j. \tag{6}$$

where $\hat{r}_{ij}$ denotes the predicted rating. As shown in Fig. 3, we test three methods on two datasets, and EMCDR -w/FL achieves a significant improvement on both metrics that verifies the effectiveness of traditional CDR methods in our scenario. Although precise matching through user-embedding can ensure the maximum transfer of common knowledge for the shared user, directly exposing user-embedding to clients in federated learning severely violates user privacy and is undesirable.

## 4.2 Why does Differential Privacy Fail to Work?

To better protect user privacy, we try to adopt differential privacy to the uploaded user embedding. First, we conduct experiments with the LDP technique on MovieLens-100K in Table 1. As expected, with a smaller privacy budget, user privacy can be guaranteed and when the privacy budget exceeds a certain threshold, the effectiveness of privacy information protection will no longer be achieved. However, due to the relatively small number of parameters in the embedding, when perturbation is applied to the model parameters to effectively protect user information, the recommendation performance decreases significantly. Achieving a balance between user privacy and recommendation performance is challenging. To address this issue, we next propose a novel FedRS framework with enhanced privacy and satisfying performance.

## 4.3 FR-JVE: Secure Preference-based Framework

To better balance the privacy and recommendation performance, we avoid directly exposing the user embedding and construct a secure rating preference for each client, serving as the transferable collaborative filtering signals to match similar users rather than the user embedding. Intuitively, a user's rating preference is suitable to represent the main characteristics of that user. For example, in [44], the rating preference of the user is the weighted sum of item embedding that is rated by this user. To ensure user privacy, here we apply model inversion techniques to distill the rating preference from both the user and item information for each client. Furthermore, we observe that the data format of each user and item is indifferentiable one-hot representation, impeding the optimization of the model inversion training. The framework of our proposed method is illustrated in Fig. 2. Consequently, we first apply the normal distribution and softmax function to initialize the data format for model inversion. Suppose that the user $u_k^i$ and item $v_k^j$ in client $k$ is recognized as :

$$u_k^i = (\mathbf{u}_1, \mathbf{u}_2, \cdots, \mathbf{u}_{|U_k|}), \;\; v_k^j = (\mathbf{v}_1, \mathbf{v}_2, \cdots, \mathbf{v}_{|V_k|}). \tag{7}$$

where $\mathbf{u}_i, \mathbf{v}_j \in \{0, 1\}$. Then we transform the one-hot representation with normal distribution and softmax function:

$$\mathbf{u}_i = \frac{exp(\hat{\mathbf{u}}_i)}{\sum_{i=1}^{|U_k|} exp(\hat{\mathbf{u}}_i)}, \quad \mathbf{v}_j = \frac{exp(\hat{\mathbf{v}}_j)}{\sum_{j=1}^{|V_k|} exp(\hat{\mathbf{v}}_j)}, \text{where } \hat{\mathbf{u}}_i, \hat{\mathbf{v}}_j \sim \mathcal{U}(0, 1). \tag{8}$$

With the new format defined in Eq.(8), each client can distill the rating preference for the user $u_k^i$ with the model $f_k$:

$$(p_k^i | u_k^i, r_i) = \min_{v_k} \mathcal{L}(f_k(u_k^i, v_k), r_i). \tag{9}$$

During the training process, the gradient of $u_k^i$ is frozen. Each element of the rating preference $p_k^i = (p_1, p_2, \cdots, p_{|V_k|})$ is a weight for the whole item embedding $V_k$. Then, the client $k$ will train the matrix factorization with the new data format like Eq.(1) to obtain the item embedding. It may raise concerns that an individual user might overlook numerous items. However, many other users involved in the federated training have rated these neglected items. As a result, these ratings contribute to well-generalized item embeddings. The overall item embedding embodies enriched knowledge, thereby facilitating the informative user characteristics. Inspired by the attention mechanism, which enables each component to contribute distinctively when compressing various elements into a singular representation, we obtain the final rating preference $P_k^i$ for the user $u_k^i$ as follows:

$$P_k^i = \sum_{j=1}^{|V_k|} p_j v_{k,e}^j. \tag{10}$$

In this work, the rating preference will be used to transfer the common knowledge from other clients, which acts as a collaborative filtering signal enhanced with secure user information. Like Eq.(5), each client locally trains a bridge function to map the aggregated rating preferences from other clients to the local rating preference.

However, simply employing the rating preference cannot match similar users like Eq.(6). Here we search the top-k similar users with the rating preference for better prediction:

$$S_k^i = top\_k\left(|r - \min_{u_{k,e}^i}[u_{k,e}^i]^T \phi_k(P_{k,t}^i)|\right). \tag{11}$$

Where $S_k^i$ is the matrix where each column is a searched user embedding, $K$ and $r$ are hyper-parameters that denote the number of collected users and the expected recommendation score, $\phi(\cdot)$ is a bridge function defined in Eq.(5) to map the rating preference here. Then, the final prediction for the user $u_k^i$ will be implemented like Eq.(10). The workflow and privacy analysis of FR-JVE are provided in Appendix A and B.

## 5 Experiments

### 5.1 Datasets & Baselines

A thorough experimental study has been conducted to assess the performance of the FR-JVE in two popular scenarios with four recommendation datasets: (1) **Rating Prediction**: MovieLens-100K (ML-100K)[7] and MovieLens-1M (ML-1M). (2) **Top-K Recommendation**: LastFM-2K(LastFM) [3] and QB-article [41]. The details of these datasets are outlined in Appendix D.

For a fair comparison with other works, we follow the same protocols proposed by [29] to stimulate FL settings. We evaluate our method with the following baselines. **1. User-based Federated Recommendation:** FedMF [4], FedNCF [10], PFedRec [27], FedRAP [21]; **2. Customed Methods:** FedMF [4]-w/DP, EMCDR [28]-w/FL+DP; **3. Federated Cross-Domain Recommendation:** FedCDR [30], P2FCDR [5]. More details are provided in Appendix C.

### 5.2 Configurations

Unless otherwise mentioned, to build the system with partially overlapped users and items, we first assign each client with random private users and items and then use the Dirichlet distribution

Table 2: Performance comparison of various methods for top-k recommendation.

| Categories | Methods | Metrics | LastFM-2K | | | QB-article-10000 | | | QB-article | | |
|---|---|---|---|---|---|---|---|---|---|---|---|
| | | | @10 | @20 | @50 | @10 | @20 | @50 | @10 | @20 | @50 |
| Customed Methods | FedMF-w/DP | HR | 0.1954 | 0.2725 | 0.4907 | 0.2140 | 0.3102 | 0.5020 | 0.2350 | 0.3188 | 0.4971 |
| | | NDCG | 0.1176 | 0.1369 | 0.1797 | 0.1263 | 0.1505 | 0.1883 | 0.1350 | 0.1561 | 0.1910 |
| | EMCDR | HR | 0.3162 | 0.4085 | 0.6087 | 0.3773 | 0.4724 | 0.6421 | 0.4667 | 0.5523 | 0.7029 |
| | | NDCG | 0.2146 | 0.2378 | 0.2770 | 0.2315 | 0.2555 | 0.2889 | 0.2996 | 0.3212 | 0.3509 |
| | EMCDR-w/FL+DP | HR | 0.4721 | 0.5501 | 0.7232 | 0.5426 | 0.6405 | 0.7899 | 0.6997 | 0.7686 | 0.8685 |
| | | NDCG | 0.3536 | 0.3732 | 0.4072 | 0.3316 | 0.3563 | 0.3859 | 0.4411 | 0.4586 | 0.4783 |
| Federated CDR | FedCDR | HR | 0.2839 | 0.4073 | 0.5890 | 0.4243 | 0.5387 | 0.6845 | 0.2891 | 0.3858 | 0.7666 |
| | | NDCG | 0.1614 | 0.1924 | 0.2284 | 0.2388 | 0.2678 | 0.2967 | 0.1756 | 0.1998 | 0.2745 |
| | P2FCDR | HR | 0.3466 | 0.4620 | 0.6808 | 0.3389 | 0.4338 | 0.6133 | 0.3690 | 0.4712 | 0.6502 |
| | | NDCG | 0.2182 | 0.2472 | 0.2902 | 0.1981 | 0.2221 | 0.2572 | 0.2168 | 0.2426 | 0.2778 |
| User-Based FedRS | FedMF | HR | 0.2087 | 0.2940 | 0.4963 | 0.2460 | 0.3296 | 0.4960 | 0.2744 | 0.3506 | 0.4953 |
| | | NDCG | 0.1256 | 0.1470 | 0.1865 | 0.1428 | 0.1638 | 0.1965 | 0.1652 | 0.1843 | 0.2127 |
| | FedNCF | HR | 0.3115 | 0.4394 | 0.6723 | 0.3291 | 0.4755 | 0.7176 | 0.4230 | 0.5588 | 0.7844 |
| | | NDCG | 0.1958 | 0.2278 | 0.2736 | 0.1825 | 0.2191 | 0.2668 | 0.2445 | 0.2787 | 0.3234 |
| | PFedRec | HR | 0.5818 | 0.6404 | 0.7779 | 0.6355 | 0.6952 | 0.8123 | 0.7066 | 0.7478 | 0.8851 |
| | | NDCG | 0.4247 | 0.4394 | 0.4663 | 0.3922 | 0.4074 | 0.4303 | 0.4583 | 0.4687 | 0.4954 |
| | FedRAP | HR | 0.6010 | 0.6528 | 0.7524 | 0.7325 | 0.8022 | 0.8311 | 0.8276 | 0.8951 | 0.9115 |
| | | NDCG | 0.4358 | 0.4489 | 0.4683 | 0.4320 | 0.4501 | 0.4558 | 0.4999 | 0.5175 | 0.5207 |
| | FR-JVE (Ours) | HR | **0.6578** | **0.6866** | **0.8093** | **0.7517** | **0.8490** | **0.8909** | **0.8635** | **0.9148** | **0.9421** |
| | | NDCG | **0.4684** | **0.4882** | **0.5022** | **0.4524** | **0.4871** | **0.4950** | **0.5141** | **0.5421** | **0.5633** |

Table 3: Performance comparison of various methods for rating prediction.

| Dataset | Metric | Customed Methods | | | Federated CDR | | User-based Federated Recommendation | | | | |
|---|---|---|---|---|---|---|---|---|---|---|---|
| | | FedMF-w/DP | EMCDR | EMCDR-w/FL+DP | FedCDR | P2FCDR | FedMF | FedNCF | PFedRec | FedRAP | FR-JVE |
| MovieLens-100K | MAE | 4.7998 | 2.1779 | 1.2693 | 2.0535 | 1.4178 | 3.6019 | 1.6982 | 1.2503 | 1.3248 | **1.1765** |
| | RMSE | 7.1237 | 2.6074 | 2.0796 | 2.6883 | 1.7954 | 4.3780 | 2.1153 | 1.6465 | 1.6725 | **1.4725** |
| MovieLens-1M-3000 | MAE | 5.2519 | 1.7794 | 1.2596 | 2.3624 | 1.2620 | 3.7927 | 1.6276 | 1.2060 | 1.1260 | **1.0496** |
| | RMSE | 10.8743 | 2.1641 | 2.7304 | 3.1643 | 1.6493 | 4.5506 | 2.0689 | 1.5565 | 1.4253 | **1.2554** |
| MovieLens-1M | MAE | 4.3850 | 1.5118 | 1.0869 | 1.9701 | 1.1593 | 3.6809 | 1.4435 | 1.1689 | 1.0912 | **1.0150** |
| | RMSE | 6.9106 | 1.8612 | 2.1763 | 2.6428 | 1.4699 | 4.3610 | 1.8457 | 1.5021 | 1.3722 | **1.2210** |

Dir($\alpha = 10$) [32] to distribute shared users to yield data heterogeneity for all datasets where a smaller $\alpha$ indicates higher data heterogeneity. Here, we report the Mean Absolute Error (MAE) and Root Mean Square Error (RMSE) as the metrics for the rating prediction [28] and Hit Rate (HR@K) and Normalized Discounted Cumulative Gain (NDCG@K) for the top-k recommendation [27, 21]. In this work, we set $K = 10$. We illustrate all the settings with all the benchmark parameters in Appendix D.

## 5.3 Performance Overview

**Main Results.** Table 2 and 3 comprehensively showcase the efficacy of various methods on two tasks. For the top-k recommendation task, when examining the HR@10 metric, FR-JVE outperforms other methods by significant margins. As the evaluation metrics extend to higher positions, FR-JVE continues to exhibit robust performance, indicating its ability to provide relevant recommendations even when considering a larger pool of candidates. Notably, PFedRec and FedRAP outperform the Federated CDR method across most datasets and achieve relatively good results due to the gain of the linear layer within their frameworks. The performance of EMCDR-w/FL+DP is closely related to the privacy budget, and the model undergoes a significant degradation under strict privacy requirements. Here, we report the case with $\epsilon = 0.2$, and the remaining analysis is provided in Table 1.

**Communication Efficiency.** Fig. 4 shows the evaluation of various methods in terms of convergence and communication efficiency. Here, we record evaluations for these methods for every two iterations. Under this joint venture ecosystem, the convergence of the FedMF method is extremely challenging, making it difficult to achieve effective recommendation performance. While FedRAP is more complex than other baselines, it needs more iterations to converge. FR-JVE achieves the best evaluations in less than 20 iterations and displays a fast convergence trend within a shorter time of 80 iterations.

**Ablation Study.** As shown in Table 5, we evaluate the effects of each module in our model via ablation studies. The -w/o bridge function denotes the performance without using the bridge function but directly employing the rating preference from other clients. The -w/o rating preference means

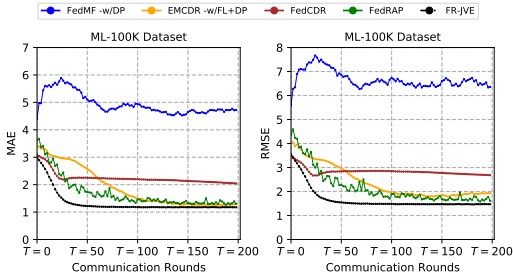
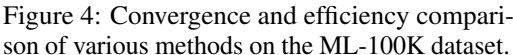

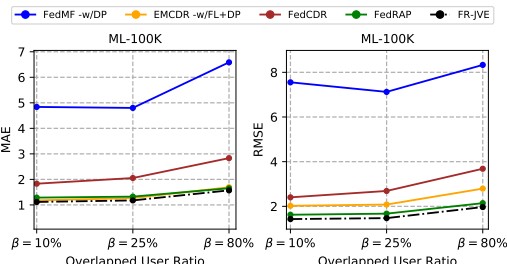

Figure 4: Convergence and efficiency comparison of various methods on the ML-100K dataset.

Figure 5: Performance comparison of various methods w.r.t. ratio $\beta$ between local unique users and shared overlapped users.

the client communicates with the only item embedding without user information. Compared with FR-JVE, the performance of FR-JVE -w/o rating preference degrades evidently. Specifically, the relatively less prominent role of the bridge function module may be constrained by issues related to the dataset, specifically that different clients did not introduce many new items or items that are sparse among clients. Experiment results verify the effectiveness of all modules.

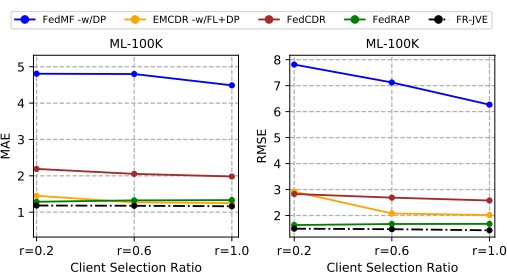

Figure 6: Performance comparison of various methods w.r.t. ratio $r$ between active clients and total clients in each round.

Table 4: Evaluation of the Top-K searching of FR-JVE in two federated recommendation scenarios. Here we underline the value used in our experiments.

| Dataset | Metric | Top-K Search | | | |
|---|---|---|---|---|---|
| | | $K = 100$ | $K = 200$ | $K = 400$ | $K = 800$ |
| ML-100K | MAE | 1.2374 | 1.1765 | 1.2248 | 1.8145 |
| | RMSE | 1.5861 | 1.4725 | 1.4995 | 2.1663 |
| | | $K = 400$ | $K = 800$ | $K = 1200$ | $K = 1600$ |
| LastFM | HR@10 | 0.6135 | 0.6394 | 0.6506 | 0.6578 |
| | NDGC@10 | 0.4211 | 0.4458 | 0.4547 | 0.4684 |

**Data Distribution.** Fig. 5 displays two metrics with different ratios between local unique users and overlapped users on MovieLens-100K. As shown in this figure, we select three different ratios to divide the dataset and experimental results show that all methods achieve an improvement with the decline in local unique users. The underlying reason may be that the more overlapped users there are, the more associations between users can be captured, and different clients can obtain useful information from other clients to improve recommendation performance. However, when the number of locally unique users is large, the differences in data distribution among different clients are relatively significant, leading to a decline in the effectiveness of federated learning.

Table 5: Ablation study with two tasks.

| Methods | MovieLens-100K | | LastFM-2K | |
|---|---|---|---|---|
| | MAE | RMSE | HR@10 | NDCG@10 |
| FR-JVE (Ours) | 1.1765 | 1.4725 | 0.6578 | 0.4684 |
| -w/o bridge function | 2.4200 | 2.7768 | 0.6281 | 0.4361 |
| -w/o rating preference | 2.9596 | 3.4988 | 0.5416 | 0.3317 |

Table 6: Time cost of Top-K searching with ML-100K (100 users, 200 items).

| Time (ms) | $K = 100$ | $K = 200$ | $K = 400$ | $K = 800$ |
|---|---|---|---|---|
| Other baselines | 9.9449 | 9.3909 | 9.6094 | 10.1382 |
| FR-JVE (Ours) | 12.4295 | 11.9481 | 12.1262 | 12.7664 |

**Parameter Sensitivity Analysis.** Fig. 6 provides the two metrics under different ratios between active clients and total clients. FR-JVE performs best with different ratios, and the lower MAE and RMSE loss is achieved by applying more active clients. Furthermore, we conduct additional research on the number of searching users $K$. When the value of $K$ is too small, the client fails to migrate sufficient knowledge from limited similar users, leading to insignificant improvement in performance. Conversely, if an excessive amount of user knowledge is migrated, it may introduce redundant information that impacts the model's judgment.

To discuss the potential extra time cost introduced by the top-k search, we conducted time tests for different k values while controlling other variables in Table 6. We assume other methods incur equal time costs with similar inference approaches. FR-JVE requires an extra step for a top-k search. The cost for "Other baselines" should theoretically be independent of the k value, but there are fluctuations. The experiments show that our top-k search does not significantly increase the cost.

Table 7: Three different bridge functions.

| FR-JVE (Bridge Function) | MovieLens-100K | | LastFM-2K | |
|---|---|---|---|---|
| | MAE | RMSE | HR@10 | NDCG@10 |
| Linear Trans. | 1.1765 | 1.4725 | 0.6578 | 0.4684 |
| Logarithmic Trans. | 1.1782 | 1.4801 | 0.6534 | 0.4609 |
| Logistic Trans. | 1.1698 | 1.4592 | 0.6488 | 0.4577 |

Table 8: Privacy enhancement with DP.

| Privacy Budget | MovieLens-100K | | LastFM-2K | |
|---|---|---|---|---|
| | MAE | RMSE | HR@10 | NDCG@10 |
| FR-JVE | 1.1765 | 1.4725 | 0.6578 | 0.4684 |
| -w/DP($\epsilon = 0.1$) | 1.1904 | 1.4918 | 0.6411 | 0.4572 |
| -w/DP($\epsilon = 0.2$) | 1.2057 | 1.4987 | 0.6251 | 0.4430 |

We use a linear function for the mapping since our experiments are conducted on standard datasets. Although real-world data may require more complex bridge functions, the results in Table 7 show limited performance gains from more complex architectures. Complex bridge functions risk overfitting, especially with sparse or noisy transferred knowledge, whereas a simple linear network effectively captures essential cross-client relationships while maintaining good generalization.

**Privacy Enhancement.** In our method, each client transfers privacy-friendly common knowledge from other clients without using the LDP algorithm, but with a distilled user's rating preference from the local dataset. Furthermore, we also provide experiments in Table 8 that apply differential privacy to our rating preferences for reference. However, our method itself can guarantee user information privacy. Considering that differential privacy can impact performance, we did not utilize it further.

Table 9: Scalability of participants.

| Methods | MovieLens-100K | | LastFM-2K | |
|---|---|---|---|---|
| | MAE | RMSE | HR@10 | NDCG@10 |
| FR-JVE ($C = 5$) | 1.1765 | 1.4725 | 0.6578 | 0.4684 |
| $C = 10$ | 1.1979 | 1.5101 | 0.6132 | 0.4152 |
| pFedRec ($C = 5$) | 1.2503 | 1.6465 | 0.5818 | 0.4247 |
| $C = 10$ | 1.2923 | 1.7524 | 0.5387 | 0.3859 |

Table 10: Add-on of Other baselines.

| Methods | MovieLens-100K | | LastFM-2K | |
|---|---|---|---|---|
| | MAE | RMSE | HR@10 | NDCG@10 |
| FedNCF | 1.6892 | 2.1153 | 0.3115 | 0.1958 |
| FR-JVE -w/FedNCF | 1.5696 | 1.9842 | 0.5950 | 0.4203 |
| pFedRec | 1.2503 | 1.6465 | 0.5818 | 0.4247 |
| FR-JVE -w/pFedRec | 1.1938 | 1.4937 | 0.6018 | 0.4409 |

**Scalability and Efficiency.** In this paper, we focus on federated recommendation within a joint-venture ecosystem, where each participant holds substantial user–item data with natural overlaps. This setting aligns with cross-silo FL, featuring few participants but complex data distributions. In contrast, cross-device FL (e.g., $C > 20$) involves numerous users with limited data, which differs from our target scenario. To validate our approach, we vary the number of participants $C$ across three configurations on two datasets. Results in Table 9 confirm the effectiveness of our method.

**Flexibility and Lightweight.** The key of FR-JVE is to allow each client to transfer more common knowledge from other clients with a distilled user's rating preference from the local dataset. Since this rating preference is unrelated to the backbone of federated recommendation algorithms, we can integrate it with other basic algorithms. In our paper, we employed the most common collaborative filtering. We will supplement the integration with well-known FedNCF and pFedRec algorithms. Experimental results 10 demonstrate that our algorithm can be combined with any basic federated recommendation algorithm to enhance performance.

# 6 Conclusion

In this paper, we delve into the challenges of developing a federated recommendation system tailored for joint venture ecosystems. We propose an innovative and privacy-enhancing framework, FR-JVE, which leverages the user's rating preferences as a filtering signal to transmit common knowledge, thereby enhancing recommendation performance while safeguarding user privacy. Extensive experiments conducted on various settings and baselines show that FR-JVE achieves significant improvement in recommendation performance.

## Acknowledgments and Disclosure of Funding

This work is supported by the National Key Research and Development Program of China under grant 2024YFC3307900; the National Natural Science Foundation of China under grants 62376103, 62302184, 62436003 and 62206102; Major Science and Technology Project of Hubei Province under grant 2024BAA008; Hubei Science and Technology Talent Service Project under grant 2024DJC078; Ant Group through CCF-Ant Research Fund; and Fundamental Research Funds for the Central Universities under grant YCJJ20252319. The computation is completed in the HPC Platform of Huazhong University of Science and Technology.

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

# A Algorithm Flow for FR-JVE

---

**Algorithm 1:** FR-JVE

---

**Input** : $T$: communication round; $K$: client number; $D_k$: local dataset for the client $k$; $u_{k,e}$: local user embedding; $v_{k,e}$: local item embedding; $\phi_k(\cdot)$: local bridge function; $f_k$: local inversion model.

**Output :** $v_e$: Item Embedding.

1 Initialize the parameter $w$;
2 **Before Federated Training:**
3 **for** *each selected client $k \in [K]$* **in parallel do**
4     Transform the local data format with (7)(8);
5     Distill the rating preference with (9)(10);
     // Upload/Download rating preference
6 **end**
7 **At Training Stage:**
8 **for** $c = 1$ **to** $T$ **do**                                         // communication round
9     Server randomly selects a subset of devices $S_t$;
10    **for** *each selected client $k \in S_t$* **in parallel do**
11       Train the local matrix factorization with new data format with (1);
12       Train the bridge function for rating preference with (5);
13       Send the item embedding $v_{k,e}$ back to the server.
14    **end**
15    $v_e \leftarrow \text{ServerAggregation}(\{v_{k,e}\}_{k \in S_t})$
16 **end**
17 **At Inference Stage:**
18 Each client matches the similar users for the shared users with (2);
19 The final prediction will be made by average rating with (6).

---

# B Privacy Analysis for FR-JVE

In our proposed FR-JVE framework, user embedding never leaves the local client. The only transferred knowledge is the rating preference and item embedding, of which the item embedding is necessary and has little user privacy, and the rating preference is the linear combination of the column vectors of item embedding $V_k$ using the elements of $p_k$ as the coefficients. However, the distillation of the rating preference employs the frozen gradient of the user information. We need to figure out whether attackers can recover user embedding information through rating preference. Assuming that attackers know their target rating, inferring the original user embedding can be understood as solving a one-line linear equation with multiple variables, where the number of variables is the dimension size of the embedding. According to Eq.(1), for the user $u_k^i$ at the client $k$, the rating is calculated as:

$$\underbrace{r_i}_{|U_k| \times |V_k|} = [\underbrace{u_{k,e}^i}_{d \times |U_k|}]^T \underbrace{\phi_k(P_{k,t}^i)}_{d \times |V_k|}.$$

It is clear that when the embedding size of users or items $d > 1$, then determining the value of each dimension in a user embedding involves solving a *multi-variable linear equation*. These variables are interrelated and cannot be solved separately, and thus, attackers fail to infer the user embedding.

# C Baselines

**1. User-based Federated Recommendation:**
 **FedMF** [4]: This approach employs matrix factorization in federated contexts to mitigate information leakage by encrypting the gradients of user and item embedding.
**FedNCF** [10]: It is the federated version of NCF. It updates user embedding locally on each client and synchronizes the item embedding on the server for global updates.
**PFedRec** [27]: It aggregates item embedding from clients on the server side and leverages them in

the evaluation to facilitate bipartite personalized recommendations.

**FedRAP** [21]: This approach further employs personalized techniques to guide the local training by gradually increasing the regularization weights to mitigate performance degradation.

**2. Customed Methods:**

**FedMF [4] - w/DP**: It conducts local matrix factorization and global aggregation of both user embedding and item embedding. For a fair comparison, the differential privacy will be only added on the uploaded user embedding.

**EMCDR [28]- w/FL+DP**: This method is a popular embedding-and-mapping framework for handling CDR. For a fair comparison, here we only map the user embedding and add the differential privacy to the uploaded user embedding.

**3. Federated Cross-Domain Recommendation:**

**FedCDR [30]**: It is a privacy-preserving federated CDR model designed for individual customer scenarios. It builds a cross-domain embedding transformation model on the server side.

**P2FCDR [5]**: It is the first privacy-preserving federated framework for dual-target cross-domain recommendation that enhances information fusion at the feature level.

## D  Datasets & Configurations

These datasets are commonly employed for evaluating recommendation systems. Specifically, two MovieLens datasets sourced from the MovieLens platform feature movie ratings spanning 1 to 5, with each user contributing at least 20 ratings. LastFM, a music recommendation dataset, comprises users' preferred artist lists and associated tags. QB-article, an implicit feedback dataset, is derived from user interaction logs.

Each experiment set is run twice, and we take each run's final 10 rounds' accuracy and calculate the average value and standard variance. We use Adam as an optimizer with a linear learning rate schedule. We set the remaining parameters according to the values in the original open-source code.

Table 11: Experimental Details. Analysis of various considered settings of different datasets in the experiments section.

| Attributes | Rating Prediction | | Top-K Recommendation | |
|---|---|---|---|---|
| | **ML-100K** | **ML-1M** | **LastFM** | **QB-article** |
| Ratings | 100,000 | 1,000,209 | 92,834 | 348,736 |
| Users | 943 | 6,040 | 1,892 | 24,516 |
| Items | 1,682 | 3,952 | 17,632 | 7,355 |
| Sparsity | $s = 93.70\%$ | $s = 95.81\%$ | $s = 99.72\%$ | $s = 99.81\%$ |
| Batch Size | $b = 32$ | $b = 64$ | $b = 256$ | $b = 256$ |
| Learning Rate | $l = 0.001$ | $l = 0.001$ | $l = 0.04$ | $l = 0.01$ |
| Local Users | 100 | 600 | 200 | 2,400 |
| Local Items | 200 | 400 | 1,700 | 700 |
| Shared Users | 443 | 3,040 | 892 | 12,516 |
| Client numbers | $C = 5$ | $C=5$ | $C=5$ | $C=5$ |
| Top-K Search | $K = 200$ | $K=3000$ | $K=1600$ | $K=7000$ |
| Local training epoch | $E = 5$ | $E = 5$ | $E = 5$ | $E = 5$ |
| Client selection ratio | $k = 0.6$ | $k = 0.6$ | $k = 0.6$ | $k = 0.6$ |
| Communication Round | $T = 200$ | $T = 200$ | $T = 200$ | $T = 200$ |

