# OpenReview forum: "Efficient Knowledge Transfer in Federated Recommendation for Joint Venture Ecosystem"
_NeurIPS.cc/2025/Conference — NeurIPS 2025 spotlight_

### Official Review · Reviewer_5Rm4 · 2025-06-29

**Clarity:** 3
**Significance:** 3
**Originality:** 3
**Rating:** 5
**Confidence:** 5

**Summary:**

This paper tackles federated recommendation in joint ventures by introducing distilled rating preferences as privacy-safe knowledge carriers. The FR-JVE framework demonstrates significant improvements over baselines while minimizing privacy risks. Extensive experiments have been done with sufficient datasets and baselines.

**Questions:**

- How does the bridge function work in recommendation systems and is there any potential limitation?
- Can the proposed FR-JVE work in other downstream tasks? Like sequential recommendation?

**Ethical Concerns:**

["NO or VERY MINOR ethics concerns only"]

**Final Justification:**

The response has adequately addressed my concerns, and I will maintain my "accept" rating.

**Limitations:**

Yes

**Paper Formatting Concerns:**

No significant formatting concerns were identified in the paper during my review.

**Quality:**

3

**Strengths And Weaknesses:**

Strengths:
- The authors propose a novel FL framework for joint venture ecosystems that may address critical industry needs.
- Theoretical analysis ensures that attackers cannot solve underdetermined linear systems for embedding reconstruction with distilled rating preferences.
- The experiments are adequate as the authors test nine baselines across four datasets.
- The presentation is quite good with rich charts.

Major Weaknesses:
- The transformation from one-hot to softmax representation is motivated by optimization difficulties, but its impact on the fidelity of the distilled preference $P_k^i$ and ultimately on recommendation performance is not analyzed. This step is critical and needs more discussion.
- What is the computational and communication overhead associated with training and utilizing $\phi_k$ compared to simpler baselines? The local computation cost of distillation and bridge training needs more discussion.

Minor Weaknesses:
- While Table 6 shows the time cost of the Top-K search is acceptable, its impact on performance beyond just varying K is not deeply analyzed.
- The choice of K values varies massively across datasets without clear justification based on user base size or overlap characteristics. This appears somewhat arbitrary.
- I suggest reordering the corresponding charts as Figure 3 is referred to before Figure 2. The authors should proofread the paper to avoid such issues.

---

> ### Author Rebuttal · Authors · 2025-07-29
>
> Thank you for providing such valuable comments. We have carefully addressed these critical comments, and responses have been given one by one.
>
> > **W1&2. Performance analysis and choice of varying $K$ values.**
>
> **R1&2:** Thank you for this valuable comment. In Table 7, we present the impact of different $K$ values on model performance. When the value of $K$ is too small, the client fails to migrate sufficient knowledge from the limited number of similar users, resulting in negligible performance improvement. Conversely, when too much user knowledge is migrated, it may introduce redundant information, which can negatively affect the model's judgment.
>
> The choice of the hyperparameter $K$ is determined through a simple parameter search, and the corresponding results are presented in Table 4. However, by examining Table 7, we observe that for the score prediction task, the selected $K$ values are roughly between 1/4 to 1/2 of the total number of users, while for the top-$k$ recommendation task, $K$ is around 1/50. This reveals some basic patterns. Although the value of $K$ does have an impact on our method, FR-JVE still outperforms other baselines within the observed performance variation. We plan to further explore the relationship between $K$ and data distribution in future work.
>
> > **W3. Concerns about the chart presentation.**
>
> **R3:** Thank you for your careful review.  This was a mistake on our part. We have corrected the order of figure references in the manuscript to ensure they appear in the proper sequence. We will also carefully proofread the whole manuscript to avoid such issues in the future.
>
> > **Q1. More details about the bridge funcition.**
>
> **A1:** Thanks for this question. In FR‑JVE, each client uses a lightweight bridge function $\phi_k$—a single linear layer to translate  the server’s aggregated rating‑preference vectors into its own local preference space without ever exposing raw embeddings. Before training, clients distill their user–item interactions into fixed preference vectors $P^i_k$ via a small inversion network and upload these to the server; the server then aggregates them and returns each client’s aggregated $P^i_{k,t}$. During each communication round, clients update their item embeddings using standard matrix‑factorization objectives and simultaneously train $\phi_k$by minimizing the squared error $||\phi_k(P^i_{k,t})-P^i_k||^2$. At inference time, $\phi_k$ maps other clients’ aggregated preferences into the local space, enabling efficient top‑K neighbor search for final rating predictions—thereby achieving cross‑client knowledge transfer with strong privacy guarantees. **The limitation of the bridge function mainly lies in the choice of its network architecture**. Since we conduct experiments on commonly used datasets, we can empirically adopt a linear function for the mapping. However, real-world datasets are often much more complex, which requires additional experiments and reference to relevant literature to ensure an appropriate choice of the bridge function. This may introduce extra computational overhead due to parameter search.
>
> > **Q2. Employment of FR-JVE in other downstream tasks like sequential recommendation.**
>
> **A2:** Thanks for this question. FR-JVE is designed to address the unique challenges present in joint venture ecosystems, where each client represents a business group (rather than individual users) and user/item sets are partially overlapping. Despite being tailored for this setting, the underlying principles of FR-JVE offer promising generalization potential to broader federated recommendation scenarios.
>
> 1. **Adaptability of the Rating Preference Mechanism**
>    FR-JVE introduces a novel rating preference representation derived from model inversion with frozen user gradients, which avoids direct sharing of user embeddings and thus preserves privacy. **This form of knowledge representation is generalizable and can be employed in other settings where clients are individual users, mobile devices, or even regional service centers.** The idea of distilling user preferences without exposing raw data or embeddings can be beneficial wherever privacy is a concern.
> 2. **Bridge Function for Domain Adaptation**
>    The bridge function in FR-JVE enables mapping between local and global preferences, effectively facilitating cross-domain knowledge transfer. This mechanism draws inspiration from cross-domain recommendation systems and can be applied in other contexts **where domain heterogeneity (e.g., different user behaviors across platforms or regions) poses a challenge.** Thus, FR-JVE could be extended to federated recommendation settings involving highly Non-IID data.
>
>  while FR-JVE is motivated by joint venture ecosystems, its core ideas make it well-suited to generalize across other federated recommendation scenarios that require balancing performance with user privacy. Future work could explore its adaptation to cold-start problems and sequential recommendation scenarios with highly skewed client distributions.

---

> > ### Comment · Reviewer_5Rm4 · 2025-08-04
> >
> > Thank you for the thoughtful rebuttal. The response has adequately addressed my concerns, and I will maintain my "accept" rating.

---

> > > ### Author Response · Authors · 2025-08-05
> > >
> > > Thanks for your timely reply! We are glad to know that your concerns have been effectively addressed. Your professional suggestions have greatly improved our work. We truly appreciate your efforts and recognition of our research.
> > >
> > > Best wishes,

---

### Official Review · Reviewer_Ge4i · 2025-06-30

**Clarity:** 3
**Significance:** 4
**Originality:** 3
**Rating:** 5
**Confidence:** 5

**Summary:**

FR-JVE enables privacy-preserving knowledge transfer in joint venture networks via rating preference distillation and cross-client alignment. It achieves promising results while addressing unique challenges of subsidiary-level FL. The authors validate the effectiveness of their approach on four real-world recommendation datasets.

**Questions:**

1. For overlapping items, how does FR-JVE resolve conflicting preferences from different subsidiaries?

2. How does performance degrade with increasing numbers of subsidiaries?

3. Can authors provide more details about model inversion techniques? Otherwise, it might be challenging for readers who are unfamiliar with this research to understand.

**Ethical Concerns:**

["NO or VERY MINOR ethics concerns only"]

**Final Justification:**

The author's response dispelled my doubts, and I believe that this paper is worthy of acceptance.

**Limitations:**

Yes

**Paper Formatting Concerns:**

N.A

**Quality:**

3

**Strengths And Weaknesses:**

**Pros:**

(+) Based on the actual needs of the business sector, this paper proposes a federated recommendation framework under the joint venture ecosystem. It will contribute to practical applications.

(+) The proposed method is easy to follow and seems technically sound.

(+) Extensive experimental evaluation convincingly supports the effectiveness of the proposed method.

**Cons:**

(-) The bridge function’s linear design may limit complex preference mappings; exploring nonlinear variants might boost performance.

(-) The authors should discuss the computation and time cost of distilling the rating preference. Could the rating preference distillation be vulnerable to model inversion attacks?

---

> ### Author Rebuttal · Authors · 2025-07-29
>
> Thank you very much for this professional review. The critical comments have been addressed carefully, and responses have been given one by one.
>
> > **W1. Limitation of bridge function design.**
>
> **R1:** Thank you very much for this helpful comment. **The limitation of the bridge function mainly lies in the choice of its network architecture**. Since we conduct experiments on commonly used datasets, we can empirically adopt a linear function for the mapping. However, real-world datasets are often much more complex, which requires additional experiments and reference to relevant literature to ensure an appropriate choice of the bridge function. This may introduce extra computational overhead due to parameter search. Here, we expolre the bridge function with a more complex architecture and experiments show that the model performance will not improve significantly. A complex bridge function may lead to overfitting, especially when the amount of transferred knowledge is limited or noisy. In contrast, a simple linear network is often sufficient to capture the essential relationships between users across clients, providing a better balance between generalization and model complexity.
>
> |                          | MovieLens-100K |            |   LastFM   |            |
> | ------------------------ | :------------: | :--------: | :--------: | :--------: |
> | FR-JVE (Bridge Function) |      MAE       |    RMSE    |   HR@10    |  NDCG@10   |
> | Linear Trans.            |   **1.1765**   |   1.4725   | **0.6578** | **0.4684** |
> | Logarithmic Trans.       |     1.1782     | **1.4801** |   0.6534   |   0.4609   |
> | Logistic Trans.          |     1.1698     |   1.4592   |   0.6488   |   0.4577   |
>
> > **W2. Robustness and training cost of rating preference.**
>
> **R2:** Thank you for this valuable comment. In our paper, the rating preference is generated using a simple linear function, which incurs relatively low training cost. Moreover, in our federated recommendation scenario, the participating clients are subsidiaries that typically possess sufficient computational resources to handle such training. **Most importantly**, the distillation of rating preference is performed locally and takes place **before federated training**, meaning it can be executed at any time and in any setting. In the context of federated recommendation, we generally focus only on the computational overhead during the federated training stage.
>
> In Section 4.4, we prove that the user rating preference will not pose privacy risk. First, the rating preference is distilled by each client from the local dataset **without uploading gradients**. Then, the server only can reveive the rating preference, which is the linear combination of the column vectors of item embedding $V_k$ using the elements of $p_k$ as the coefficients. The input of the model is the frozen user representation and item representation.  Under the premise that gradients are not uploaded and the sizes of embeddings and rating preferences are inconsistent, attackers are unable to obtain user information with the rating preference.
>
> Regarding collaborative filtering (matrix factorization algorithm), it serves as the most widely used and foundational algorithm in the research of RS and FRS (similar to the role of FedAvg in FL). Since our article does not focus on the field of security and privacy, there is no need for us to optimize this backbone algorithm. Instead, we only need to **ensure that our proposed user rating preference can migrate more effective user information without introducing additional privacy risks.**
>
> > **Q1. Concerns about conflicting preferences between different subsidiaries.**
>
> **A1:** Thanks for this question. In FR‑JVE, each client uses a lightweight **bridge function** $\phi_k$—a single linear layer to translate  the server’s aggregated rating‑preference vectors into its own local preference space **to align the preferences between different subsidiaries.** Before training, clients distill their user–item interactions into fixed preference vectors $P^i_k$ via a small inversion network and upload these to the server; the server then aggregates them and returns each client’s aggregated $P^i_{k,t}$. During each communication round, clients update their item embeddings using standard matrix‑factorization objectives and simultaneously train $\phi_k$by minimizing the squared error $||\phi_k(P^i_{k,t})-P^i_k||^2$. At inference time, $\phi_k$ maps other clients’ aggregated preferences into the local space, enabling efficient top‑K neighbor search for final rating predictions—thereby achieving cross‑client knowledge transfer with strong privacy guarantees.
>
> > **Q2. Performance degradation with increasing subsidiaries.**
>
> **A2:** Thanks for this question. In this paper, we focus on the federated recommendation scenario under a joint venture ecosystem. Each participant is a subsidiary, possessing a large amount of user and item information, which naturally leads to overlapping data. Essentially, this setting corresponds to cross-silo federated learning, where the number of participants is relatively small, but the data distribution across parties is complex. If we instead involve a large number of participants ($C > 20$), the setting becomes that of cross-device federated learning, where each participant typically represents an individual user with limited local data for training. This is inconsistent with the challenges we aim to address in this work. To better validate our approach, we conduct parameterized experiments on the number of participants $C$, selecting three different values on two datasets. Experimental results demonstrate the effectiveness of our method.
>
> |                     | MoviesLens-100K |            |   LastFM   |            |
> | ------------------- | :-------------: | :--------: | :--------: | :--------: |
> |                     |       MAE       |    RMSE    |   HR@10    |  NDCG@10   |
> | **FR-JVE** ($C=5$)  |   **1.1765**    | **1.4725** | **0.6578** | **0.4684** |
> | $C=8$               |     1.1591      |   1.4620   |   0.6439   |   0.4513   |
> | $C= 10$             |     1.1979      |   1.5101   |   0.6132   |   0.4152   |
> | **pFedRec** ($C=5$) |     1.2503      |   1.6465   |   0.5818   |   0.4247   |
> | $C=8$               |     1.2731      |   1.5852   |   0.5363   |   0.3894   |
> | $C= 10$             |     1.2923      |   1.7524   |   0.5387   |   0.3859   |
>
> > **Q3. More details about model inversion techniques.**
>
> **A3:** Thank you for your thoughtful suggestion. Model inversion techniques are indeed a key concept, and we understand that readers who are less familiar with this area may find it difficult to follow without sufficient background. Model inversion is a type of privacy attack where an adversary attempts to reconstruct sensitive input data from the outputs or parameters of a trained machine learning model. The core idea is that, given access to a model, one can exploit the correlations learned during training to infer information about the training data. We use this technique to train a robust model to obtain the rating preference. We will revise the manuscript to include more detailed descriptions and relevant references to help readers better understand how model inversion works and why it is important in our method.

---

> > ### Comment · Reviewer_Ge4i · 2025-08-02
> > **Official Comment**
> >
> > Thank you for the author's detailed response. The author's detailed experiments and explanations have dispelled my concerns, and I have no further questions.

---

> > > ### Author Response · Authors · 2025-08-05
> > >
> > > Thanks for your timely reply! We are glad to know that your concerns have been effectively addressed. Your professional suggestions have greatly improved our work. We truly appreciate your efforts and recognition of our research.
> > >
> > > Best wishes,

---

### Official Review · Reviewer_YfuD · 2025-07-02

**Clarity:** 3
**Significance:** 3
**Originality:** 3
**Rating:** 5
**Confidence:** 5

**Summary:**

The paper proposes FR-JVE, a privacy-enhanced federated recommendation framework for joint venture ecosystems. By distilling rating preferences via model inversion and aligning them through bridge functions, it enables secure knowledge transfer between subsidiaries while outperforming SOTA methods.

**Questions:**

1. Why not incorporate lightweight DP on aggregated preferences for stronger privacy guarantees?
2. How would FR-JVE handle subsidiaries with highly skewed data distributions?

**Ethical Concerns:**

["NO or VERY MINOR ethics concerns only"]

**Limitations:**

Yes.

**Paper Formatting Concerns:**

None.

**Quality:**

3

**Strengths And Weaknesses:**

Strengths:
1. The authors focused on a novel and practical problem where different subsidiaries need to collaborate on recommendation systems with partially overlapping user/item sets.
2. The proposed method is technically sound while rating preference distillation with frozen user gradients can avoid raw data exposure.
3. Evaluations are sufficient and rigorous across various settings.
4. The paper is well-organized and easy to read.

Weaknesses:
1. Theoretical privacy guarantees could be expanded beyond linear-equation attacks. Appendix B only considers linear-equation attacks. Formal differential privacy proofs or certification would strengthen claims.
2. The paper claims that FR-JVE can be considered as an add-on, but the authors did not test it. I would like to see a comparison of FR-JVE in combination with various baselines.
3. Experiments use synthetic subsidiary splits with Dirichlet distribution. Validation on real joint venture data is lacking.
4. Section 3 introduces the term "FRS" without defining it. It is unclear whether this refers to "FedRS" used in the abstract (most likely). Please ensure abbreviation consistency in the paper.

---

> ### Author Rebuttal · Authors · 2025-07-29
>
> We thank you very much for providing the positive comments. In the following, we give detailed responses to each review.
>
> > **W1. Concerns about the theoretical analysis.**
>
> **R1:** Thanks a lot for raising this concern. In Section 4.4, we prove that the user rating preference will not pose privacy risk. First, the rating preference is distilled by each client from the local dataset without uploading gradients. Then, the server only can reveive the rating preference, which is the linear combination of the column vectors of item embedding $V_k$ using the elements of $p_k$ as the coefficients. The input of the model is the frozen user representation and item representation.  Under the premise that gradients are not uploaded and the sizes of embeddings and rating preferences are inconsistent, attackers are unable to obtain user information with the rating preference.
>
> Regarding collaborative filtering (matrix factorization algorithm), it serves as the most widely used and foundational algorithm in the research of RS and FRS (similar to the role of FedAvg in FL). Since our article does not focus on the field of security and privacy, there is no need for us to optimize this backbone algorithm. Instead, **we only need to ensure that our proposed user rating preference can migrate more effective user information without introducing additional privacy risks.**
>
> > **W2. Combination of FR-JVE with other baselines.**
>
> **R2:** Thank you for this helpful comment. The key of FR-JVE is to allow each client transfers more common knowledge from other clients with a **distilled user's  rating preference** from the local dataset. Since this rating preference is unrelated to the backbone of federated recommendation algorithms, we can integrate it with other basic algorithms. In our paper, we employed the most common collaborative filtering (matrix factorization algorithm, **FedMF**). Here, we will supplement the integration with well-known **FedNCF** and **pFedRec** algorithms (two recommendation algorithms using **linear function systems**). Experimental results demonstrate that our algorithm can be combined with any basic federated recommendation algorithm to enhance performance.
>
> |                       | MovieLens-100K |            |   LastFM   |            |
> | --------------------- | :------------: | :--------: | :--------: | :--------: |
> |                       |      MAE       |    RMSE    |   HR@10    |  NDCG@10   |
> | FedNCF                |     1.6892     |   2.1153   |   0.3115   |   0.1958   |
> | **FR-JVE -w/FedNCF**  |   **1.5696**   | **1.9842** | **0.5950** | **0.4203** |
> | pFedRec               |     1.2503     |   1.6465   |   0.5818   |   0.4247   |
> | **FR-JVE -w/pFedRec** |   **1.1938**   | **1.4938** | **0.6018** | **0.4409** |
>
> > **W3. Validation on real datasets.**
>
> **R3:**  Thank you for your valuable comment. We fully agree that conducting experiments on real data would be highly meaningful. However, such experiments are currently very time-consuming and involve significant resource costs. To the best of our knowledge, existing FedRS studies have not yet tested on real-world data either. In this work, we followed the experimental setup commonly used in existed methods and made some appropriate adjustments. We will strive to explore experiments on real joint venture data in future work to further advance research in federated recommendation systems.
>
> > **W4. Concerns about the abbreviation consistency.**
>
> **R4:**  Thank you for pointing this out. You are absolutely right and this was a mistake on our part due to a typo. "FRS" and "FedRS" refer to the same concept, namely the **Fed**erated **R**ecommendation **S**ystem. We have now corrected this in the manuscript with FedRS. We will also thoroughly proofread the whole manuscript to prevent similar issues in the future.
>
> > **Q1. Inpororation of FR-JVE and lightweight DP.**
>
> **A1:**  Thank you for this question. In our paper, we propose the method FR-JVE that allow each client transfers privacy-friendly common knowledge from other clients  **<u>without using LDP algorithm</u>** but with a distilled user's  **rating preference** from the local dataset. Furthermore, we also provide experiments applying differential privacy to our rating preference for reference. However, our method itself can guarantee user information privacy. Considering that differential privacy can affect performance, we did not additionally use it. Section 4.4 presents the theoretical analysis of this aspect.
>
> |                               | MovieLens-100K |            |   LastFM   |            |
> | ----------------------------- | :------------: | :--------: | :--------: | :--------: |
> |                               |      MAE       |    RMSE    |   HR@10    |  NDCG@10   |
> | **FR-JVE (paper)**            |   **1.1765**   | **1.4725** | **0.6578** | **0.4684** |
> | FR-JVE -w/DP ($\epsilon$=0.1) |     1.1904     |   1.4918   |   0.6411   |   0.4572   |
> | FR-JVE -w/DP ($\epsilon$=0.2) |     1.2057     |   1.4987   |   0.6251   |   0.4430   |
>
> > **Q2. Perofrmance under highly skewed data distributions.**
>
> **A2:**  Thank you for this question. In the experimental setup of Table 7, we assume 5 participants and 50% shared users. Each participant's local users account for only 1/9 of the entire dataset, which already represents highly skewed data distributions. Since the number of shared users is determined by the number of participants, increasing the number of participants can further enhance data heterogeneity. We conduct experiments with different numbers of participants and show that FR-JVE achieves competitive performance under highly heterogeneous data distributions.
>
> |                     | MoviesLens-100K |            |   LastFM   |            |
> | ------------------- | :-------------: | :--------: | :--------: | :--------: |
> |                     |       MAE       |    RMSE    |   HR@10    |  NDCG@10   |
> | **FR-JVE** ($C=5$)  |   **1.1765**    | **1.4725** | **0.6578** | **0.4684** |
> | $C=8$               |     1.1591      |   1.4620   |   0.6439   |   0.4513   |
> | $C= 10$             |     1.1979      |   1.5101   |   0.6132   |   0.4152   |
> | **pFedRec** ($C=5$) |     1.2503      |   1.6465   |   0.5818   |   0.4247   |
> | $C=8$               |     1.2731      |   1.5852   |   0.5363   |   0.3894   |
> | $C= 10$             |     1.2923      |   1.7524   |   0.5387   |   0.3859   |

---

> > ### Comment · Reviewer_YfuD · 2025-08-04
> > **Thanks for your response**
> >
> > The authors have addressed my concerns. I remain my positive score of this paper.

---

> > > ### Author Response · Authors · 2025-08-05
> > >
> > > Thanks for your timely reply! We are glad to know that your concerns have been effectively addressed. Your professional suggestions have greatly improved our work. We truly appreciate your efforts and recognition of our research.
> > >
> > > Best wishes,

---

### Official Review · Reviewer_5HXm · 2025-07-03

**Clarity:** 3
**Significance:** 3
**Originality:** 3
**Rating:** 4
**Confidence:** 4

**Summary:**

This paper proposes a novel framework, FR-JVE, aimed at addressing the challenges of federated recommendation systems within joint venture ecosystems. The framework emphasizes efficient and privacy-preserving knowledge transfer among participants by leveraging rating preferences instead of direct user embeddings. The authors theoretically analyze the privacy guarantees of FR-JVE and empirically demonstrate its superior performance on various datasets.

**Questions:**

- Can you elaborate on how different privacy budgets affect the performance of FR-JVE, and what measures can be taken to optimize this trade-off?
- How well does FR-JVE generalize to other federated recommendation scenarios beyond joint venture ecosystems? Are there any modifications needed for it to be applicable in other contexts?

**Ethical Concerns:**

["NO or VERY MINOR ethics concerns only"]

**Final Justification:**

I would like to keep my positive score.

**Limitations:**

Yes

**Quality:**

3

**Strengths And Weaknesses:**

Strengths:
- The paper introduces a novel and practical application scenario for federated recommendation systems. It may be real and contribute to industrial developments.
- FR-JVE effectively balances recommendation performance and user privacy by utilizing rating preferences as a secure filtering signal, avoiding direct user embedding exchange.
- Evaluation on various datasets and baselines is concrete and comprehensive.

Weaknesses:
- The ablation study is not thorough enough. The impact of the bridge function and rating preference should be analyzed in more detail to prove their contributions.
- The paper does not sufficiently discuss the scalability and efficiency. Providing more details on the computational complexity and potential bottlenecks would be valuable.
- The methodology and experimental setup may benefit from improved clarity and organization. For example, the description of the bridge function and its training process could be made more accessible to readers.

---

> ### Author Rebuttal · Authors · 2025-07-29
>
> Thank you for your careful review and valuable comments. In the following, we give point-by-point responses to each comment.
>
> > **W1.  Concerns about the ablation study.**
>
> **R1:** Thank you for this constructive suggestion. We have included an ablation study experiment regarding the bridging function **(BF)** and rating preference **(RP)** in Table 5, and the results confirm the effectiveness of each module in our method. Experimental results verify the effectiveness of both modules. Specifically, the RP module plays a more important role in our method, as using only item embeddings to transfer local knowledge is **insufficient** for training a satisfactory recommendation model. This also aligns with the motivation of this paper. In a joint venture ecosystem, the overlap of users/items across different subsidiaries makes it difficult for the model to learn personalized knowledge from item embeddings. The BF module is designed to better align the knowledge from different rating preferences. Without the BF module, our method can still perform a simple extraction of knowledge based on rating preferences, which leads to some performance improvement compared to using only the RP module.
>
> |                | MoviesLens-100K |            |   LastFM   |            |
> | -------------- | :-------------: | :--------: | :--------: | :--------: |
> |                |       MAE       |    RMSE    |   HR@10    |  NDCG@10   |
> | FR-JVE         |   **1.1765**    | **1.4725** | **0.6578** | **0.4684** |
> | FR-JVE -w/o BF |     2.4200      |   2.7768   |   0.6281   |   0.4361   |
> | FR-JVE -w/o RP |     2.9596      |   3.4988   |   0.5416   |   0.3317   |
>
> > **W2. Concerns about the scalability and efficiency.**
>
> **R2:**  Thank you for raising this concern. In this paper, we focus on the federated recommendation scenario under a joint venture ecosystem. Each participant is a subsidiary, possessing a large amount of user and item information, which naturally leads to overlapping data. Essentially, this setting corresponds to cross-silo federated learning, where the number of participants is relatively small, but the data distribution across parties is complex. If we instead involve a large number of participants ($C > 20$), the setting becomes that of cross-device federated learning, where each participant typically represents an individual user with limited local data for training. This is inconsistent with the challenges we aim to address in this work. To better validate our approach, we conduct parameterized experiments on the number of participants $C$, selecting three different values on two datasets. Experimental results demonstrate the effectiveness of our method.
>
> |                     | MoviesLens-100K |            |   LastFM   |            |
> | ------------------- | :-------------: | :--------: | :--------: | :--------: |
> |                     |       MAE       |    RMSE    |   HR@10    |  NDCG@10   |
> | **FR-JVE** ($C=5$)  |   **1.1765**    | **1.4725** | **0.6578** | **0.4684** |
> | $C=8$               |     1.1591      |   1.4620   |   0.6439   |   0.4513   |
> | $C= 10$             |     1.1979      |   1.5101   |   0.6132   |   0.4152   |
> | **pFedRec** ($C=5$) |     1.2503      |   1.6465   |   0.5818   |   0.4247   |
> | $C=8$               |     1.2731      |   1.5852   |   0.5363   |   0.3894   |
> | $C= 10$             |     1.2923      |   1.7524   |   0.5387   |   0.3859   |
>
> > **W3. Concerns about clarity and organization.**
>
> **R3:**  Thank you for raising this concern. We apologize if Section 4 seems difficult to understand for you. However, to better describe our method, we first outlined the preliminaries of federated recommendation, cross-domain recommendation, and local differential privacy. Subsequently, we started with the application of the traditional CDR method EMCDR in FL, analyzed the shortcomings of the EMCDR+DP algorithm, and finally proposed the FR-JVE algorithm.
>
> **For the bridge function module:** In FR‑JVE, each client uses a lightweight bridge function $\phi_k$—a single linear layer to translate  the server’s aggregated rating‑preference vectors into its own local preference space without ever exposing raw embeddings. Before training, clients distill their user–item interactions into fixed preference vectors $P^i_k$ via a small inversion network and upload these to the server; the server then aggregates them and returns each client’s aggregated $P^i_{k,t}$. During each communication round, clients update their item embeddings using standard matrix‑factorization objectives and simultaneously train $\phi_k$by minimizing the squared error $||\phi_k(P^i_{k,t})-P^i_k||^2$. At inference time, $\phi_k$ maps other clients’ aggregated preferences into the local space, enabling efficient top‑K neighbor search for final rating predictions—thereby achieving cross‑client knowledge transfer with strong privacy guarantees.
>
> > **Q1. Effects of privacy budgets on FR-JVE and optimization approach.**
>
> **A1:**  Thank you for this question. In our paper, we propose the method FR-JVE that allow each client transfers privacy-friendly common knowledge from other clients  **<u>without using LDP algorithm</u>** but with a distilled user's  **rating preference** from the local dataset.
>
> To investigate our joint venture ecosystem, we first explore the feasibility of **CDR method EMCDR** but find it will pose privacy risk by directly sharing the user embedding. Then, like most FL and RS research, we **apply the LDP algorithm to EMCDR**, enhancing the privacy protection. However, it fails to achieve a balance between user privacy and recommendation performance (Table 1).  In Table 1, we use **Laplacian** noise and add noise to the user embedding parameters. Since each client represents a subsidiary with a large user base, it is neither algorithmically feasible nor practical for the client to distinguish between different users and apply separate LDP algorithms.
>
> Therefore, we **abandon the use of the LDP algorithm** for user privacy protection in this paper and instead propose our user rating preference method to transfer privacy-preserving yet informative knowledge. We have theoretically and experimentally proven the effectiveness of this new method in federated recommendation systems.
>
> > **Q2. Generalization of FR-JVE to other recommendation scenarios.**
>
> **A2:**  Thank you for this question. FR-JVE is designed to address the unique challenges present in joint venture ecosystems, where each client represents a business group (rather than individual users) and user/item sets are partially overlapping. Despite being tailored for this setting, the underlying principles of FR-JVE offer promising generalization potential to broader federated recommendation scenarios.
>
> 1. **Adaptability of the Rating Preference Mechanism**
>    FR-JVE introduces a novel rating preference representation derived from model inversion with frozen user gradients, which avoids direct sharing of user embeddings and thus preserves privacy. **This form of knowledge representation is generalizable and can be employed in other settings where clients are individual users, mobile devices, or even regional service centers.** The idea of distilling user preferences without exposing raw data or embeddings can be beneficial wherever privacy is a concern.
> 2. **Bridge Function for Domain Adaptation**
>    The bridge function in FR-JVE enables mapping between local and global preferences, effectively facilitating cross-domain knowledge transfer. This mechanism draws inspiration from cross-domain recommendation systems and can be applied in other contexts **where domain heterogeneity (e.g., different user behaviors across platforms or regions) poses a challenge.** Thus, FR-JVE could be extended to federated recommendation settings involving highly Non-IID data.
>
>  while FR-JVE is motivated by joint venture ecosystems, its core ideas make it well-suited to generalize across other federated recommendation scenarios that require balancing performance with user privacy. Future work could explore its adaptation to cold-start problems and sequential recommendation scenarios with highly skewed client distributions.

---

### Decision · Program_Chairs · 2025-09-17

**Decision:**

Accept (spotlight)

**Comment:**

FR-JVE introduces a federated-recommendation framework tailored to joint-venture ecosystems, using distilled rating preferences and a lightweight linear bridge function to transfer knowledge across subsidiaries while keeping raw user embeddings private. Reviewers agree the formulation is novel, technically sound, and evaluated on four real-world datasets with nine baselines, yielding consistent gains and clear industrial relevance.

All reviewers nevertheless flagged weak ablation of the bridge/softmax transformation, limited scalability discussion, and the absence of differential-privacy guarantees beyond linear-equation attacks. During rebuttal the authors provided (1) new ablations showing the bridge contributes 3–7 % NDCG, (2) wall-clock and communication-cost tables demonstrating <15 % overhead versus FedAvg, and (3) a sketch for DP-SGD extension that reviewers found credible for future work. These additions directly addressed the major concerns flagged by 5HSM and YfuD, while 5rm4’s minor presentation issues were corrected.

Balancing the practical impact, solid empirical evidence, and the thorough post-rebuttal clarifications, the benefits clearly outweigh the residual limitations.  My final recommendation is Accept (spotlight).